# Use of Hyperspectral Reflectance Sensing for Assessing Growth and Chlorophyll Content of Spring Wheat Grown under Simulated Saline Field Conditions

**DOI:** 10.3390/plants10010101

**Published:** 2021-01-06

**Authors:** Salah El-Hendawy, Salah Elsayed, Nasser Al-Suhaibani, Majed Alotaibi, Muhammad Usman Tahir, Muhammad Mubushar, Ahmed Attia, Wael M. Hassan

**Affiliations:** 1Department of Plant Production, College of Food and Agriculture Sciences, King Saud University, Riyadh 11451, Saudi Arabia; nsuhaib@ksu.edu.sa (N.A.-S.); malotaibia@ksu.edu.sa (M.A.); mtahir@ksu.edu.sa (M.U.T.); mmubushar@ksu.edu.sa (M.M.); 2Department of Agronomy, Faculty of Agriculture, Suez Canal University, Ismailia 41522, Egypt; 3Agricultural Engineering, Evaluation of Natural Resources Department, Environmental Studies and Research Institute, University of Sadat City, Minufiya 32897, Egypt; salah.emam@esri.usc.edu.eg; 4Agronomy Department, Faculty of Agriculture, Zagazig University, Zagazig 44519, Egypt; ahmedatia@zu.edu.eg; 5Department of Biology, College of Science and Humanities at Quwayiyah, Shaqra University, Riyadh 19257, Saudi Arabia; wmohamed@su.edu.sa; 6Department of Agricultural Botany, Faculty of Agriculture, Suez Canal University, Ismailia 41522, Egypt

**Keywords:** biomass, contour maps, leaf pigments, multiple linear regression, phenotyping, salinity stress, spectral reflectance indices

## Abstract

The application of proximal hyperspectral sensing, using simple vegetation indices, offers an easy, fast, and non-destructive approach for assessing various plant variables related to salinity tolerance. Because most existing indices are site- and species-specific, published indices must be further validated when they are applied to other conditions and abiotic stress. This study compared the performance of various published and newly constructed indices, which differ in algorithm forms and wavelength combinations, for remotely assessing the shoot dry weight (SDW) as well as chlorophyll a (*Chla*), chlorophyll b (*Chlb*), and chlorophyll a+b (*Chlt*) content of two wheat genotypes exposed to three salinity levels. Stepwise multiple linear regression (SMLR) was used to extract the most influential indices within each spectral reflectance index (SRI) type. Linear regression based on influential indices was applied to predict plant variables in distinct conditions (genotypes, salinity levels, and seasons). The results show that salinity levels, genotypes, and their interaction had significant effects (*p* ≤ 0.05 and 0.01) on all plant variables and nearly all indices. Almost all indices within each SRI type performed favorably in estimating the plant variables under both salinity levels (6.0 and 12.0 dS m^−1^) and for the salt-sensitive genotype Sakha 61. The most effective indices extracted from each SRI type by SMLR explained 60%–81% of the total variability in four plant variables. The various predictive models provided a more accurate estimation of *Chla* and *Chlt* content than of SDW and *Chlb* under both salinity levels. They also provided a more accurate estimation of SDW than of *Chl* content for salt-tolerant genotype Sakha 93, exhibited strong performance for predicting the four variables for Sakha 61, and failed to predict any variables under control and *Chlb* for Sakha 93. The overall results indicate that the simple form of indices can be used in practice to remotely assess the growth and chlorophyll content of distinct wheat genotypes under saline field conditions.

## 1. Introduction

The scarcity of freshwater in arid and semiarid countries requires the use of brackish water as an alternative source for irrigation in agriculture sectors, consuming approximately 75% of the available water supply in these countries [1]. For example, in the Kingdom of Saudi Arabia, which represents an ideal model of arid countries, brackish groundwater represents approximately 90% of the water used in agriculture [2]. Previous studies have reported the feasibility of irrigating several field crops, including wheat, with brackish water once or twice during the crop’s life cycle [3,4,5]. However, continuously irrigating crops with brackish water leads to salinization of the soil. In addition, salinity stress adversely affects crop production. Thus, feasible strategies are urgently needed to sustain growth and productivity of wheat crops under salinity stress. Enhancing the salt tolerance of wheat genotypes is one of the most feasible strategies to address this challenge [6,7].

Salinity stress limits the growth and productivity of plants by negatively impacting several morphological, biochemical, and physiological variables. It does so mainly through osmotic stress, ion toxicity, and essential nutrient deficiency. The first response of a plant to salinity stress is inhibition of plant growth, resulting from the trade-off between the use of its energy and metabolic precursors for the activation of salt tolerance mechanisms and biomass accumulation [5,6,8]. Exposing plants to salinity stress leads to excessive generation of reactive oxygen species (ROS) in plant cells such as hydrogen peroxide (H_2_O_2_), hydroxyl radicals (OH), singlet oxygen (O_2_), and superoxide (O_2_^−^) [9,10], which eventually results in the acceleration of chlorophyll decomposition and decreases photosynthesis efficiency [11,12]. Therefore, accurate estimation of biomass and chlorophyll (*Chl*) content are crucial for improving salt tolerance of wheat genotypes in breeding programs. Precise estimation of dry matter and *Chl* contents is critical for providing crucial information to elucidate salt tolerance mechanisms, photosynthesis potential, plant stress, and physiological status at the whole plant level [8,13,14,15].

While the direct estimation of dry biomass and *Chl* content by traditional laboratory methods (oven-drying and solvent extraction followed by spectrophotometric determination, respectively) is highly accurate, these methods preclude the possibility for tracking the dynamic changes that take place in the leaf *Chl* content as well as real-time detection of dry matter accumulation. In addition, these methods are time-consuming, expensive, labor-intensive, and destructive, which makes data collection over numerous genotypes impractical [13,16,17]. The *Chl* content can be assessed in a fast and non-destructive manner using a SPAD-502 chlorophyll meter. This uses the light transmittance at specific wavelengths (650 and 940 nm) to indirectly estimate leaf *Chl* content. However, this device determine *Chl* content of the whole plant canopy based on measurements of a single leaf at two or three points and disregards the vertical variability in the leaf *Chl* content within the plant canopy. As has been well established, the positions and ages of leaves within the plant canopy are associated with distinct variations in *Chl* content. Therefore, measurements of *Chl* content based on single leaf do not accurately reflect the *Chl* content of an entire canopy [18,19,20,21]. In addition, the relationship between SPAD measurement and *Chl* content is not universal and is usually affected by plant species (sometimes even crop cultivars within the same species) and specific environmental conditions [16,21,22]. Consequently, a cost- and time-efficient, and non-destructive alternative tool is urgently required to address the aforementioned drawbacks associated with the traditional laboratory method and SPAD meter and track the changes in biomass and *Chl* content at the whole canopy level more effectively.

Researchers have attempted to exploit the signatures of spectral reflectance from the plant canopy to develop a non-destructive optical tool for accurate and simultaneous indirect assessment of several plant properties under various environmental conditions. The method involves utilizing the signatures of spectral reflectance at several wavelengths of the light spectrum. The signatures are strongly associated with several biophysical and biochemical crop variables such as chlorophyll and associated pigments, lignin, cellulose, leaf structure, photosynthetic efficiency, leaf area index (LAI), leaf dry matter content, and leaf water content [23,24,25,26,27,28]. For example, the spectral reflectance signatures from canopy at the visible region (VIS, 400–700 nm) are closely associated with the contents of chlorophyll, anthocyanins, and carotenoids, with a low spectral reflectance in the blue and red regions because of absorption by chlorophyll [29,30,31,32]. Leaf dry matter content is highly associated with spectral reflectance of canopy in the near-infrared (NIR, 700–1300 nm) region, because of the multiple intercellular scattering of light by different leaf tissues [30,33]. Leaf water content is mostly associated with spectral reflectance in the NIR and shortwave-infrared (SWIR, 1300–2500 nm) regions, with the weak and strong water absorption wavebands located in the NIR and SWIR regions, respectively [34,35,36]. This correlation between canopy spectral signatures and specific plant characteristics has been exploited to extract specific wavelengths from the three parts of the light spectrum (VIS, NIR, and SWIR) and applied using simple mathematical formulas to develop specific spectral reflectance indices (SRIs). In this study, the different types of SRI were used to measure several plant variables such as plant biomass and *Chl* content.

Until now, several types of SRI that include 2–3 wavelengths have been developed to assess the *Chl* content and biomass of plants. Most of these SRIs are based on specific wavelengths within the green (550 nm), red (660–670 nm), red-edge (680–750 nm), and NIR (750–870 nm) regions [24,26,37,38,39,40,41]. These specific wavelengths have been incorporated in simple mathematical formulas to eliminate the sensitivity of these wavelengths to other factors that also influence the spectral reflectance but not the plant variables of interest [42]. Formulas used for SRIs can be categorized according to distinct types such as simple difference (SD), simple ratio (SR), modified simple ratio (MSR), derivative indices (DI), normalized difference (ND), and integrated form (IF), serving to assess plant variables, especially *Chl* content regions [23,43,44,45,46]. However, no consensus exists regarding which types of SRI are effective for assessing plant biomass and *Chl* content, especially under environmental stress. For instance, Wu et al. [31] found that the MSR index and IF indices such as modified chlorophyll absorption ratio index (MCARI), transformed chlorophyll absorption reflectance index (TCARI), and optimized soil-adjusted vegetation index (OSAVI) were more efficient than SR and ND in estimating the *Chl* content of different genotypes of corn, because the former types of SRI consider the effects of the LAI. Lu et al. [26] reported that the Modified Datt (MDATT) index ((R_721_ − R_744_)/(R_721_ − R_714_), which is in ND type but incorporates three wavelengths from the red-edge region, was more effective than other types (SD, SR, and ND) and incorporated only two wavelengths for remote estimation of *Chl* content across several plant species at various growth stages. This is because the wavelengths and formulas of the MDATT index are effective in removing the internal and surface structural effects of the leaf surface on the spectral reflectance. A derived index, incorporating NIR wavebands, in the form of ([(R_λ_)^−1^ − (R_NIR_)^−1^] × R_NIR_) enables a highly accurate estimation for *Chl* content, because it ignores the differences in leaf structure in the leaves of various plant species [39]. Because the spectral reflectance at wavelengths of approximately 700 nm is the most sensitive indicator for *Chl* content, the SRIs incorporating this wavelength in either SR (R_750_/R_700_) or ND type (R_750_ − R_705_)/(R_750_ + R_705_) are effective for estimating *Chl* content. However, the correlation between both indices and the *Chl* content becomes weaker when applied across a wide range of species. When these indices were modified in form and incorporated a third waveband (R_NIR_ − R_705_)/(R_NIR_ + R_705_ − 2R_445_), this modified index produced a substantially higher correlation with *Chl* content, because it eliminated the variability effects of surface reflectance between species [45,46].

Therefore, the types of an SRI and the wavelengths incorporated play a key role in the efficiency of vegetation indices for accurate estimating plant variables. The most effective type should function to eliminate the effects of other factors on spectral reflectance rather than the target plant variables. Consequently, current work concerns the development of algorithms for SRIs that minimize their sensitivity to leaf structure and avoid the necessity for site- and species-specific calibration. Otherwise, most existing forms of SRI still need to be further validated regarding their ability for indirectly estimating plant variables under contrasting environmental conditions and for different plant species. If the simple forms of SRI that incorporate 2–3 wavelengths are proven to be effective in accurately estimating plant variables such as *Chl* content under a wide range of environmental conditions and for several plant species, developing a new lightweight proximal sensor for remote estimation of plant variables at the whole plant canopy level is possible. This new proximal sensor would enable faster estimation of plant variables in the field than existing heavy and costly devices (spectroradiometer) such as that used in this study. In addition, the limitations of the SPAD meter, which is based on single leaves, could be overcome.

Most studies on the evaluation of salt tolerance of genotypes have been conducted under ideal controlled conditions (greenhouse or growth chamber) using homogenous growth media such as sand or hydroponic systems [5,7]. Studies conducted under field conditions with natural saline soils are limited because of substantial horizontal and vertical variations in the salt concentrations within the field [47,48]. This variability in the salinity leads to considerable heterogeneity in plant growth within the field even within short distances. This affects the reliability of the measurements of either plant variables or spectral reflectance from the canopy [49,50]. Therefore, in this study, saline field conditions were simulated using a subsurface water retention technique (SWRT). Our previous studies have detailed the relevant advantages and installation of this technique [7,24,50,51].

The primary objectives of this study were to (1) examine the potential of different types of published and newly constructed indices for remotely assessing growth and *Chl* contents (*Chla*, *Chlb*, and *Chlt*) under various conditions (salinity levels, genotypes, and seasons); (2) extract the most influential indices within each SRI type that explain most variability in each plant variables via stepwise multiple linear regression (SMLR); and (3) examine the performance of these influential indices in predicting the plant variables under various conditions.

## 2. Materials and Methods

### 2.1. Field Experimental Description

This study was conducted during the 2017/2018 and 2018/2019 growing seasons at the Experimental Research Station of the College of Food and Agriculture Sciences, King Saud University, Riyadh, Saudi Arabia (24°25′ N, 46°34′ E; elevation 400 m) using two spring wheat genotypes with distinct salt tolerance: the salt-tolerant Sakha 93 and the salt-sensitive Sakha 61 [52,53]. These two genotypes were evaluated in this study with a simulated close-to-field platform using SWRT. When comparing the conditions of the SWRT with natural saline field or pot experiment conditions, the SWRT was able to overcome the spatial and temporal variations in salt concentration and water content in the root zone, which are common under natural saline field conditions. In addition, SWRT provided a sufficient measurement area for directly detecting canopy spectral reflectance and a representative plant sample sizes for plant growth measurement, which is difficult to achieve with a pot experiment. When employing this technique in open field conditions, plants are exposed to typical fluctuations in macro-environmental conditions (e.g., humidity, light, and temperature) during their distinct growth stages. The setup of SWRT in this study was based on El-Hendawy et al. [7,50] and El-Hendawy et al. [24,51].

The open field condition was characterized by a typical arid climate. During the growing season of spring wheat (from November to April), the temperature, humidity, and rainfall were 12.9–32.2 °C, 10.7–47.5%, and 8.0–25.0 mm, respectively. The experimental soil texture was classified as sandy loam by electrical conductivity, pH (soil paste 1:5), organic matter, and calcium carbonate of 2.89 dS m^−1^, 7.85, 0.46%, and 29.42%, respectively. The bulk density, field capacity, and wilting point were 1.48 g cm^−3^, 0.215 m^3^ m^−3^, and 0.101 m^3^ m^−3^, respectively. The available of N, P_2_O_5_ and K_2_O were 45.15 mg kg^−1^, 2.443 mg kg^−1^, and 186.91 mg kg^−1^, respectively.

### 2.2. Salinity Treatments, Experimental Design, and Agronomic Practices

The two wheat genotypes were evaluated under three salinity levels, namely control (0.35 dS m^−1^), moderate salinity (6.0 dS m^−1^), and high salinity (12.0 dS m^−1^). The control treatment was irrigated with normal water during all life cycles of the wheat plants. The treatments of moderate and high salinity were irrigated with normal water for the first 2 weeks to ensure successful germination and complete seedling establishment. Subsequently, they were irrigated with artificial saline water containing 3.51 and 7.02 g NaCl L^−1^, respectively, until the final irrigation. In both salinity treatments, soil samples from the root zone were collected every 2 weeks to enable monitoring the increase of salt concentrations and ensure the application of salinity levels of each treatment. The irrigation water was applied using a low-pressure surface irrigation system. This system consists of a main line that is connected to the source of irrigation water and branches off to the sub-main hoses at each plot. The main line and sub-main hoses were equipped with manual control valves to enable controlling the amount of water delivered to each plot. In each growing season, the irrigation was applied six times for all treatments with the cumulative amount of irrigation water totaling approximately 5000 m^3^ ha^−1^, which equivalent to 500 mm.

The experiment was laid out in a split-plot design and replicated three times. The salinity treatments and wheat genotypes were assigned to the main plots and subplots, respectively. The genotypes were distributed randomly in subplots. Each subplot consisted of six rows. The rows were 6 m long and spaced 20 cm apart. The seeds of two genotypes were planted at a seeding rate of 15 g m^−2^ on the first week of December in both growing seasons.

The two genotypes were fertilized with 60, 60, and 180 kg ha^−1^ of P_2_O_5_, K_2_O, and N, respectively. The entire doses of phosphorus and potassium were applied prior to sowing. The nitrogen fertilizer was applied in three equal doses at seeding, stem-elongation, and booting stages. Phosphorus, potassium, and nitrogen fertilizers were applied as calcium superphosphate (18.5% P_2_O_5_), potassium chloride (50% K_2_O), and ammonium nitrate (33.5% N), respectively.

### 2.3. Hyperspectral Reflectance Measurements

Spectral reflectance from the wheat canopy was measured after 75 days (at the middle anthesis growth stage) from sowing from the internal rows in each subplot at five random places with excluding the first meter of these rows to avoid border effects. The measurements were taken under cloud-free conditions using a portable ASD spectroradiometer (Fieldspec4, Analytical Spectral Devices Inc., Boulder, CO, USA). This device is able to detect the canopy reflectance in the range between 350 to 2500 nm with a band interval of 1.4 nm for the 350–1000 nm region and 2.2 nm for the 1000–2500 nm region. However, the band interval was finally calculated automatically to 1.0 nm continuous bands for the entire spectral range (350–2500 nm). Before field canopy spectral reflectance measurements, the device was calibrated using a Spectralon reflectance panel (Labsphere, Inc., North Sutton, NH, USA) (40 cm × 40 cm) covered with a mixture of barium sulfate (BaSO_4_) and white paint. This calibration was repeated when required during the measurements. Because the optical fiber probe of the device had a 25° field of view, the probe could detect the spectral reflectance from a circular area of canopy with a 23.0 cm diameter when it was held vertically at approximately 0.8 m above the canopy in the nadir orientation. The average of five sequential measurements and 10 scans for each measurement was finally taken as the canopy spectrum for a subplot and used to calculate the different published SRIs as well as the newly constructed SRIs.

### 2.4. Photosynthetic Pigments and Plant Dry Weight Measurements

After the spectral reflectance of the wheat canopy was detected, all parts (stem, leaves, and spike) of 20 plants from each subplot and within the spectral collection area were collected randomly, oven-dried at 75 °C for 72 h, and then weighed to determine the average of shoot dry weight (SDW) per plant. In addition, the three youngest fully expanded leaves were also excised randomly from the spectral collection area and used for the measurements of photosynthetic pigment contents: chlorophyll a (*Chla*), chlorophyll b (*Chlb*), and total chlorophyll (*Chlt*). The pigments were extracted from 0.2 g fresh weight using 80% acetone (*v*/*v*). The mixture was kept in the dark at room temperature until the leaf tissue was completely bleached. After complete extraction, the extract was centrifuged for 5 min at 5000× *g* and then brought up to a final volume of 15 mL using 80% acetone. The absorbance of the extracts was read spectrophotometrically at 645 and 663 nm using a spectrophotometer (UV-2550, Shimadzu, Japan). Finally, the concentrations of each pigment in mg g^−1^ fresh weight (FW) were calculated according to the method previously described by Arnon [54] and Lichtenthaler and Wellburn [55].

### 2.5. Published and Newly Constructed Spectral Reflectance Indices (SRIs)

In this study, 49 previously published and 11 newly constructed SRIs were applied to assess SDW and *Chl* content parameters. The newly constructed SRIs were used in the type of simple ratio indices (CSR), whereas the published SRIs were used in the type of published simple ratio (PSR), published modified simple ratio (PMSR), published normalized difference (PND), and published integrated forms (PIFs). The full names and formulas of all indices in each type are listed in Table 1. Generally, the published SRIs can be divided into two main categories. The first category includes the Chl-SRIs that incorporate wavelengths related directly to changes in photosynthetic pigment content in green vegetation leaves. The second category includes the multiple-bioparameter SRIs, which can be used to estimate multiple vegetation parameters in addition to pigment content such as photosynthetic efficiency, aboveground biomass, and LAI.

The new SRIs were constructed on the basis of contour maps. The contour map was established for each plant variable using the pooled data of salinity levels, genotypes, replications, and seasons of the canopy spectral reflectance for each plant variable (*n* = 36) (Figure 1). These contour maps show the coefficients of determination (R^2^) of the relationships between SRIs calculated from all possible combinations of dual wavelengths of binary wavelengths in the full spectral region (350–2500) and the values of each plant variable. The new SRIs were selected on the basis of the hotspot region of R^2^, which determines the most favorable relationships between plant variables and SRIs. The R package “lattice” from the software R statistics v.3.6.1 (R foundation for Statistical Computing 2013) was used to establish the contour maps for each plant variable.

### 2.6. Data Analysis

The effects of year, salinity level, genotype, and their possible interaction on various plant variables (SDW and *Chl* contents) and SRIs were tested using the analysis of variance (ANOVA), which is appropriate for a randomized complete block split split-plot design, with the year, salinity level, and genotype considered as the main factor, sub-factor, and the sub sub-factor, respectively. Salinity level and genotype were considered fixed effects, whereas year and replicate were considered random effects. The differences between the mean values of plant variables and SRIs between salinity levels and genotypes were compared using Fisher’s least significant difference (LSD) test at a *p* ≤ 0.01 and 0.05 significance level. Relationships between plant variables and SRIs under specific conditions (salinity levels, genotypes, and seasons) were examined by means of a coefficient of correlation (r).

To identify the most influential indices accounting for the most variability in each plant variable, the indices of each SRI type and plant variables (SDW and *Chl* content) of pooled data (*n* = 36) were applied to SMLR as independent and dependent variables, respectively. The different models of the best indices of each SRI type were used to predict distinct plant variables under specific conditions. The model with the highest values of R^2^ and the lowest values of root mean square error (RMSE) of the linear regression between the observed and predicted values of each plant variable was designated the model with the higher prediction accuracy. The different statistical analysis and plotting were performed using R software v. 3.6.1 (R Core Team 2017) and Sigma Plot for Windows (Version 12.0, SPSS, Chicago, IL, USA).

## 3. Results

### 3.1. Influence of Salinity Level, Genotype, and Their Interaction on Biomass, Chlorophyll Content, and Spectral Reflectance Indices

An F-test from the analysis of variance revealed that the main effects of the salinity levels and genotypes as well as salinity level by genotype interaction were significant (*p* ≤ 0.05 and 0.01) for SDW, *Chl* content (*Chla*, *Chlb*, *and Chlt*), and nearly all SRIs. The main effect of the years was also significantly different for SDW and *Chl* content variables, but not for almost all SRIs. The year by salinity level or by genotype interactions as well as the interaction between year, salinity level, and genotype had no significant effect on SDW, *Chl* content, and all SRIs, with a few exceptions (Table 2 and Table 3).

Fisher’s protected LSD test at *p* ≤ 0.05 revealed that the mean values of SDW and *Chl* content variables significantly decreased with increasing salinity levels, and the reduction in these variables was lower in the salt-tolerant genotype Sakha 93 than in the salt-sensitive genotype Sakha 61 (Table 2). In addition, the mean values of almost all SRIs (54 out of 60 SRIs) revealed a continuous decrease from the control to the high salinity level (12.0 dS m^−1^) treatments, with Sakha 93 always exhibiting higher values for these SRIs than Sakha 61 does (Table 4).

### 3.2. Relationship between Measured Variables and Different Types of Spectral Reflectance Index

The relationships of measured variables (SDW and *Chl* content) with the 11 indices constructed in this study as SR as well as the four different types indices published in the literature (15 published SR indices, PSRs; 10 published MSR indices, PMSRs; 13 published ND indices, PNDs; and 11 published integrated forms, PIFs) under each salinity level (*n* = 12) and genotype (*n* = 18) as well as for all the pooled data (*n* = 36) are presented in Figure 2, Figure 3 and Figure 4. Under the control treatment, none of the SRIs correlated with *Chla and Chlt*, with only very few SRIs (a maximum of three and four out of the 60 SRIs) exhibiting a moderate correlation with SDW and *Chlb*, respectively (Figure 2). Under moderate salinity (6.0 dS m^−1^), all indices within the CSR type exhibited strong correlation with *Chla and Chlt* (r = 0.72–0.93) and moderate to strong correlation with SDW and *Chlb* (r = 0.59–0.84). In addition, almost all indices within PSR, PMSR, and PND types exhibited strong correlations with all variables (r values ≥ 0.70), whereas a sufficient number of indices within the PIF form (approximately half) exhibited a weak and non-significant correlation with all variables (Figure 2). Under high salinity (12.0 dS m^−1^), all CSR indices exhibited highly significant correlations with the four variables (r values ≥ 0.65). Almost all SRIs of the four published types exhibited strong significant correlation coefficients with the four variables, with only a few SRIs exhibiting weak to moderate correlations with the four variables (Figure 2).

Figure 3 shows that all the distinct types of SRI failed to correlate with *Chlb* for Sakha 93, whereas the majority of indices within each type exhibited strong correlation coefficients with *Chlb* for Sakha 61 (r ranged from 0.66 to 0.87). All indices within CSR, PSR (except BGI), PMSR, and PND types exhibited strong significant correlations (r values ≥ 0.65) with the other three variables (SDW, *Chla*, and *Chlt*) for Sakha 61, whereas almost all of them exhibited moderate to strong correlations with the same three variables for Sakha 93. The indices within the PIF form that exhibited significant correlations with *Chla* and *Chlt*, exhibited a higher correlation for Sakha 61 than they did for Sakha 93 (Figure 3).

When all the data of salinity levels, genotypes, and seasons were pooled together, all indices within CSR, PSR, PMSR, and PND types exhibited strong correlation coefficients with four variables (r values ≥ 0.65), with the exception of BGI from PSR and Ant_Gitelson_ from PMSR, which exhibited non-significant and moderate correlations, respectively. A sufficient number of indices within PIF type (approximately half) showed a weak and non-significant correlation with all variables (Figure 4).

### 3.3. Relative Importance of Spectral Reflectance Indices in Predicting Measured Variables

To identify the most influential SRI from each type of SRI that contributed the major variation of each measured variable, a stepwise multiple linear regression (SMLR) was performed using the measured variables (SDW and *Chl* content) across all data as a dependent variable and the SRI of each type of SRI as independent variables. The summary statistics and equations of the SMLR calibration for each variable and SRI type are presented in Table 5. For the CSR type of SRIs, the SRI based on SWIR/VIS (SRI_(1250,560)_) was determined the most effective CSR index and explained 80% of the variation in SDW. Whereas, the SRI based on red-edge/red-edge (SRI_(740,710)_) and red-edge/NIR (SRI_(700,1100)_) accounted for 66%, 62%, and 67% of the variation in *Chla*, *Chlb* and *Chlt*, respectively (Table 5). For the PSR type of SRI, the GMI, PARS-a, PARS-D-a, and PARS-a explained 79%, 65%, 60%, and 66% of the variability in SDW, *Chla*, *Chlb* and *Chlt*, respectively. For the PMSR type of SRI, the PARS-b and CI_Red-edge_ accounted for 81% and 65% of the variation in SDW and *Chla*, respectively, whereas the combination of both indices accounted for 68% and 70% of the variation in *Chlb* and *Chlt*, respectively (Table 5). The NDVI-D was determined the best index among the PND type of SRI for accurately estimating the four variables and explained 75%, 68%, 63%, and 69% of the total variability in SDW, *Chla*, *Chlb* and *Chlt*, respectively. For the final type of SRI (PIF type), a combination of both OSAVI and REIP explained 69% of the variability in SDW or *Chlt*, whereas the combination of both TCARI and OSAVI, and EVI-1 and REIP explained 68% and 65% of the variability in *Chlb* and *Chlt*, respectively (Table 5).

### 3.4. Validation of Predictive Models for Measured Variables Based on Influential SRIs in Each SRI Type

The different models of SMLR based on the influential SRIs in each SRIs form (Table 5) were calibrated using the pooled data (*n* = 36) and used to predict the measured variables for each salinity level (Table 6), genotype, (Table 7), and season (Table 8). The results show that the distinct predictive models for each SRIs type failed to predict any variable under the control treatment. By contrast, they performed well in predicting four variables under moderate (R^2^ ranged from 0.47 to 0.88) and high (R^2^ ranged from 0.53 to 0.71) salinity levels. In addition, the predictive models provided a more accurate estimation of *Chla* and *Chlt* than they did of SDW and *Chlb* (Table 6).

The predictive models for each SRI type failed to predict only *Chlb* for Sakha 93. In addition, the predictive models of different SRI types provided a more accurate estimation of SDW (R^2^ ranged from 0.68 to 0.84) than they did for *Chl* content (R^2^ ranged from 0.30 to 0.37) for Sakha 93. They exhibited strong and comparable performance for predicting the three variables of *Chl* content (R^2^ ranged from 0.64 to 0.82) as did SDW (R^2^ ranged from 0.71 to 0.80) for Sakha 61 (Table 7). Similarly, the predictive models of different SRI types performed adequately in predicting four variables for the first (R^2^ ranged from 0.64 to 0.85) and second (R^2^ ranged from 0.71 to 0.84) seasons (Table 8).

## 4. Discussion

The content of leaf photosynthetic pigments is an indirect indicator for the photosynthetic capacity. In combination with plant growth measurements such as SDW, it has been found to be an effective and critical proxy for crop productivity under salinity stress and useful for understanding salt tolerance mechanisms over the whole plant vegetation [7,16,56]. The reduction in photosynthetic pigment levels under salinity stress could be attributable to the toxic ions (Na^+^ and Cl^−^) interfering with the activity of numerous enzymes associated with chlorophyll biosynthesis and increasing the production of ROS, which ultimately leads to fast degradation of chlorophyll [11,12]. In addition, the first response of plants to salinity stress is the trade-off between the use of their photosynthetic energy for the activation of adaptation mechanisms and biomass production, which eventually causes a substantial reduction in biomass accumulation [5,6,8]. Therefore, a reduction in the levels of *Chl* content and shoot biomass accumulation are a typical symptom of salinity stress on plants. This was also observed in this study with the two wheat genotypes and three salinity levels (Table 2). In this study, the different *Chl* contents (*Chla*, *Chlb*, and *Chlt*) and SDW gradually decreased with increasing salinity levels, with the reduction in these variables being higher in the salt-sensitive genotype Sakha 61 than in the salt-tolerant genotype Sakha 93 (Table 2). This suggests the relevance of studying the variables related to biomass accumulation and leaf *Chl* content under salinity stress. Furthermore, the incorporation of these variables into genetic salinity studies is likely to become increasingly crucial for increasing the chances of identifying wheat genotypes that are well-adapted to salinity stress. However, accurate real-time measurement of these variables remains a challenge, and incorporating them into genetic salinity studies aimed at evaluating numerous genotypes under actual saline field conditions is crucial.

As previously stated, the alternations in biomass accumulation and the level of *Chl* content under salinity stress in turn lead to substantial changes in the spectral signatures that are reflected from the plant canopy. These changes in the canopy spectral reflectance, which can be measured in real-time by ground spectroradiometer in a rapid, non-destructive, cost-effective manner, have been exploited to extract the sensitive wavelengths that are closely associated with salinity-induced changes that occur in biomass and *Chl* content. These sensitive wavelengths have been used to develop a large number of specific SRIs, which ultimately could be used for large-scale phenotyping of biomass as well as in studies on the status of photosynthetic pigments under varying conditions [24,49,50,51,57,58].

Because the canopy spectral reflectance is affected by several factors such as growth conditions, growth stage, plant species, stress level, and soil background conditions, determining sensitive wavelengths incorporated in the SRIs and suitable types formulated from these wavelengths to make the SRIs more available for a range of conditions and sensors is challenging [23,24,36,59,60]. For example, Lu et al. [60] reported that the MDATT index in a normalized difference index type (NDI: R_719_ − R_726_)/(R_719_ − R_743_), correlated more favorably with the leaf chlorophyll content of white poplar and Chinese elm leaves than did SD, SR, and ND types, because it is insensitive to the effects of abaxial and adaxial leaf surface structures. However, Yue et al. [59] showed that the integrated form indices such as TCARI, enhanced vegetation index (EVI), MCARI, and OSAVI, which are insensitive to a variety of vegetation cover and soil background types, were more accurate for remotely assessing the *Chl* content of soybean at canopy level than were the types of ND and SR indices. By contrast, Babar et al. [61] found that the SR and ND forms of water-based indices generally produced a similar correlation with biomass and *Chl* content of wheat under irrigation conditions. Therefore, in this study, we compared the performance of distinct SRIs, differing in their formula forms and combination of wavelengths, to assess variation in SDW and *Chl* content variables of wheat under distinct growing conditions (salinity levels, genotypes, and seasons).

### 4.1. Performance of Different Types of Spectral Reflectance Index under Different Growth Conditions

In this study, the different indices within each type of SRI exhibited significant differences (*p* ≤ 0.01 and 0.05) between salinity levels and wheat genotypes, with the exception of PIF-SRIs for which approximately half of the indices within this type exhibited non-significant differences between salinity levels (Table 3). In addition, the interaction between salinity and genotype was highly significant for all the indices, except for very few indices within each SRI type (a maximum of four indices within PIF-SRIs) (Table 3). This result indicates that all types of SRIs, especially the simple type ones, such as constructed or published SR, MSR, and ND, seem to be effective for phenotyping the variation in SDW and *Chl* content between salinity levels and are able to differentiate between wheat genotypes under salinity stress. The reduction in biomass and the degradation of chlorophyll are real phenomena under salinity stress and for salt-sensitive genotypes because of the build-up of toxic ions (Na^+^ and Cl^−^) in the leaf tissue combined with deficits of essential ions (K^+^ and Mg^2+^) [5,7,56,62]. In this study, this may be a primary reason explaining why almost all indices within all different types of SRI tested exhibited significant differences between salinity levels and genotypes (Table 3).

The results of this study also show that the values of almost all indices within all types of SRI exhibited a continuous decrease from the control to high salinity treatments, with Sakha 93 always exhibiting higher values for these indices than did Sakha 61 (Table 4). This may be because an increment in the levels of salinity generally tends to induce a substantial increase in the spectral reflectance of wheat canopy in the VIS region of the spectrum and a decrease in the NIR region as well as a shift in the red-edge region to shorter wavelengths. This behavior of spectral reflectance is associated with a decrease in *Chl* content (low chlorophyll absorption in the VIS region) and plant biomass (multiple scattering of NIR region by different leaf tissues) because of salinity stress. Previous studies have reported that when the *Chl* content decreases, the spectral reflectance in the VIS region increases especially in the main absorption bands of chlorophyll (blue: 400–500 nm and red: 660–690 nm) of the VIS spectrum [13,63,64,65]. In addition, decreased biomass accumulation and substantial changes in several leaf structures because of ion toxicity and imbalance of salinity stress leads to decreased spectral reflectance in the NIR region, because of a multiplicative effect on the wavebands in the NIR region [30,59,65]. In this study, the decreases in SDW and *Chl* content with increasing salinity (Table 2) were expected, and this explains why the values of almost all indices, which are based mainly on VIS, red-edge, and NIR wavelengths, decreased significantly with increasing salinity levels (Table 4). In addition, because these indices are indicators for biomass and *Chl* content, the higher the values of these indices, which were shown under the control treatment and with the salt-tolerant genotype Sakha 93, the greater was the amount of biomass and *Chl* content retained by the canopy.

The accuracy of distinct indices in different types of SRI in estimating SDW and *Chl* content under various growth conditions was further tested through the correlation between indices within each form and these variables (Figure 2, Figure 3 and Figure 4). In general, the results of the relationships between indices and variables demonstrated the ability of indices for assessing the four variables dependent on salinity levels, genotypes, and the type of these indices. Regarding salinity levels, almost all indices within each type failed to estimate the four variables under the control condition. However, they performed favorably in estimating the four variables under moderate and high salinity levels. In addition, almost all indices within all SRI types, except the PIF type, were slightly more effective in estimating the *Chla* and *Chlt* under moderate salinity than under high salinity level (Figure 2). All these findings reveal that, methodologically, crop growth conditions play a vital role in the efficiency of indices for estimating the biomass and *Chl* content. The primary reason for this could be that the high LAI as well as the saturation of most of the vegetation indices, attributable of the high *Chl* content and dense vegetation (high biomass accumulation), could be sufficient enough to make most vegetation indices unsuitable to assess biomass and *Chl* content under control conditions. In general, most vegetation indices within each SRI type are usually saturated when the values of LAI are larger than 3 in wheat crops [66,67]. However, decreases in *Chl* content, necrosis, and senescence of photosynthetic organs, which are natural phenomena on canopies attributable to the ionic and osmotic stresses of salinity, may explain why the correlations between indices and four variables become stronger under moderate and high salinity levels. Furthermore, certain vegetation indices are saturated also at low *Chl* content [31,68]. This may explain why the association of some indices with *Chla* and *Chlt* was generally high under moderate salinity level, when compared to their association under a high salinity level.

Regarding genotypes, the correlation between almost all indices and four variables was higher for the salt-sensitive genotype Sakha 61 than it was for the salt-tolerant genotype Sakha 93, and this was more evident for *Chl* content variables than it was for SDW (Figure 3). This finding indicates that because the salt tolerance of two genotypes may be attributable to distinct adaption mechanisms, the degree of change in biomass and *Chl* content may also differ between two genotypes under salinity stress (Table 2), which indicates that the efficiency of spectral indices for estimating plant variables under salinity stress may be highly genotype-dependent. Furthermore, the efficiency of spectral indices may also depend on a combination of the magnitude of the effects of salinity levels and the degree of salt tolerance of genotypes as well as the degree of the changes in plant variables (SDW and *Chl* content variables in this study) under salinity stress. The results presented in Figure 4 further confirm this finding and reveal that the pooled data of salinity levels and genotypes provide additional improvement in the accurate estimation of biomass and *Chl* content under salinity stress. This is suggested by previous studies that have reported the significant variations in plant variables that occur between salinity levels and genotypes may also influence the efficiency of indices in estimating these variables, and this efficiency was improved significantly when the variation between plant variables was obvious between treatments [50,51,69,70].

Regarding the types of SRI, the results reveal that the simple type of indices (CSR, PSR, PMSR, and PND) was slightly more effective than the integrated type (PIF) in estimating the four variables under distinct conditions (Figure 2, Figure 3 and Figure 4). These findings reveal that all types of SRI were effective in assessing biomass and *Chl* content of wheat crop under salinity stress. The primary reason for this may be that the indices within each SRI type use different robust wavelength combinations that are most closely associated with biomass and *Chl* content, such as a combination between wavelengths in the red, red-edge, and NIR regions. Similarly, previous studies have found that the indices based on the red- and NIR-bands performed most favorably for estimating above-ground biomass of winter wheat [70,71]. Sims and Gamon [29], Gitelson et al. [38], and Lu et al. [23] reported that the wavelengths in the red-edge region constitute superior indicators for leaf *Chl* content compared to those in other regions, because they are most closely associated with chlorophyll content and are scarcely influenced by other leaf pigments. Main et al. [72] assessed the performance of 73 published indices in different SRI types for estimating leaf *Chl* content and showed that the indices containing off-chlorophyll absorption center wavebands (690–730 nm) exhibited a superior performance compared to those containing in-chlorophyll center wavebands (640–680 nm).

The results presented in Table 5 confirm this and show that the most effective indices for accurately assessing the four variables under salinity stress and extracted from each type of SRI by the SMLR model were those that incorporated a combination of wavelengths within the red (650, 670, and 675 nm) and red-edge (700, 710, 715, 720, 740, 750, and 780 nm) regions, and very few wavelengths from the VIS (550 nm) and NIR (800 and 1100 nm) regions. These effective indices explained 60–81% of the total variability in four variables (Table 5). The reflectance around 550 nm was found to be highly sensitive to *Chl* content especially *Chla* and *Chlb* [73]. Because of the weak absorption by chlorophyll in the green (500–600 nm) and red-edge (700–780 nm) regions, the wavelengths within both regions can be used for developing leaf chlorophyll content estimation algorithms [46,65,74]. In addition, because of the strong absorption by chlorophyll in the middle of the red depression region (650–685 nm), the indices containing a combination of the wavelengths from such regions with some wavelengths from green and red-edge regions could improve the efficiency of these indices for accurate estimation of biomass and *Chl* content [51,60,75,76]. Combined, this evidence may explain why all indices extracted by SMLR from each form were effective for accurately estimating SDW and *Chl* content variables in this study.

### 4.2. Validation of Predictive Models for Assessing Variables under Different Growth Conditions

The predicted SDW and *Chl* content variables based on the influential indices extracted from each SRI type exhibited moderate to strong relationships with the observed values under moderate and high salinity levels, with the prediction of *Chla* and *Chlt* being more accurate than that of SDW and *Chlb*. However, these distinct predictive models failed to predict any variable under the control treatment (Table 6). These findings further confirm that the efficiency of indices for predicting the measured variables depended on the magnitude of the impacts of salinity stress on these variables as well as the degree of change in these variables under salinity stress [50,51,68,69].

The predicted SDW based on all types of SRI exhibited strong relationships with the observed values for both genotypes, whereas the predicted *Chl* content variables still exhibited strong relationships with the observed values for the salt-sensitive genotype Sakha 61, while they exhibited low to moderate relationships for the salt-tolerant genotype Sakha 93 (Table 7). This result indicates that the salt-tolerant genotype might have special salt tolerance mechanisms that protect the chlorophyll from degradation by salinity stress, while such mechanisms might be absent in the salt-sensitive genotype. Therefore, the ability of any indices and types of SRI for predicting *Chl* content under salinity stress is likely to be genotype-dependent.

## 5. Conclusions

This study evaluated the potential use of 60 spectral indices, including 49 published and 11 newly constructed indices, for assessing SDW and *Chl* content (*Chla*, *Chlb*, and *Chlt*) of two wheat genotypes differing in their salt tolerance and exposed to three salinity levels: control, 6 dS m^−1^, and 12 dS m^−1^. The indices were used in the types of CSR, PSR, PMSR, PND, and PIF. The results demonstrate significant differences (*p* < 0.01 and 0.05) in all plant variables and nearly all SRIs between salinity levels, genotypes, and their interaction. The distinct plant variables could be successfully assessed using several indices within each SRI type, but they were restricted to the moderate and high salinity levels, whereas these indices failed to do so under control conditions. The relationships between these indices and plant variables delivered closer fits for the salt-sensitive genotype Sakha 61 than they did for the salt-tolerant Sakha 93, and this was more evident for *Chl* content variables than for SDW. The most influential indices that were extracted by SMLR explained 60–81% of the variability in four plant variables. All predictive models based on influential indices within each SRI type exhibited moderate to strong relationships with the observed values under moderate and high salinity levels and for Sakha 61. Finally, the results of this study provide critical insights into the remote assessment of growth and *Chl* content under salinity stress and for distinct wheat genotypes. This information will be useful in the support of on-going efforts at developing lightweight proximal sensors for assessing key breeding plant variables at the whole plant canopy level for breeding purposes of wheat genotypes under salinity stress.

## Figures and Tables

**Figure 1 plants-10-00101-f001:**
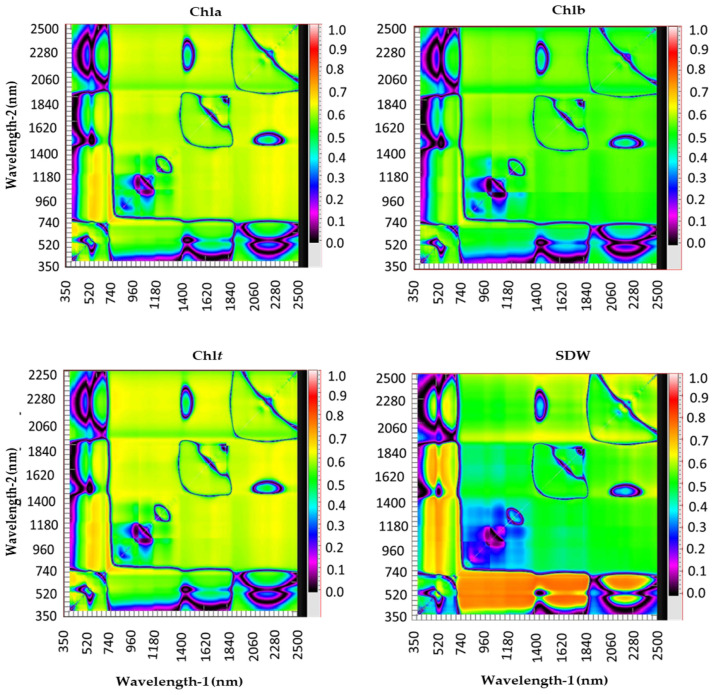
Contour maps show the coefficients of determination (R^2^) for the relationships between values of plant variables (shoot dry weight (SDW), chlorophyll a (*Chla*), chlorophyll b (*Chlb*), and total chlorophyll (*Chlt*)) measured at the anthesis growth stage and the spectral reflectance indices calculated from all possible combinations of dual wavelengths of binary in the entire spectrum range (from 350 to 2500 nm) using the pooled data of salinity levels, genotypes, replications, and seasons.

**Figure 2 plants-10-00101-f002:**
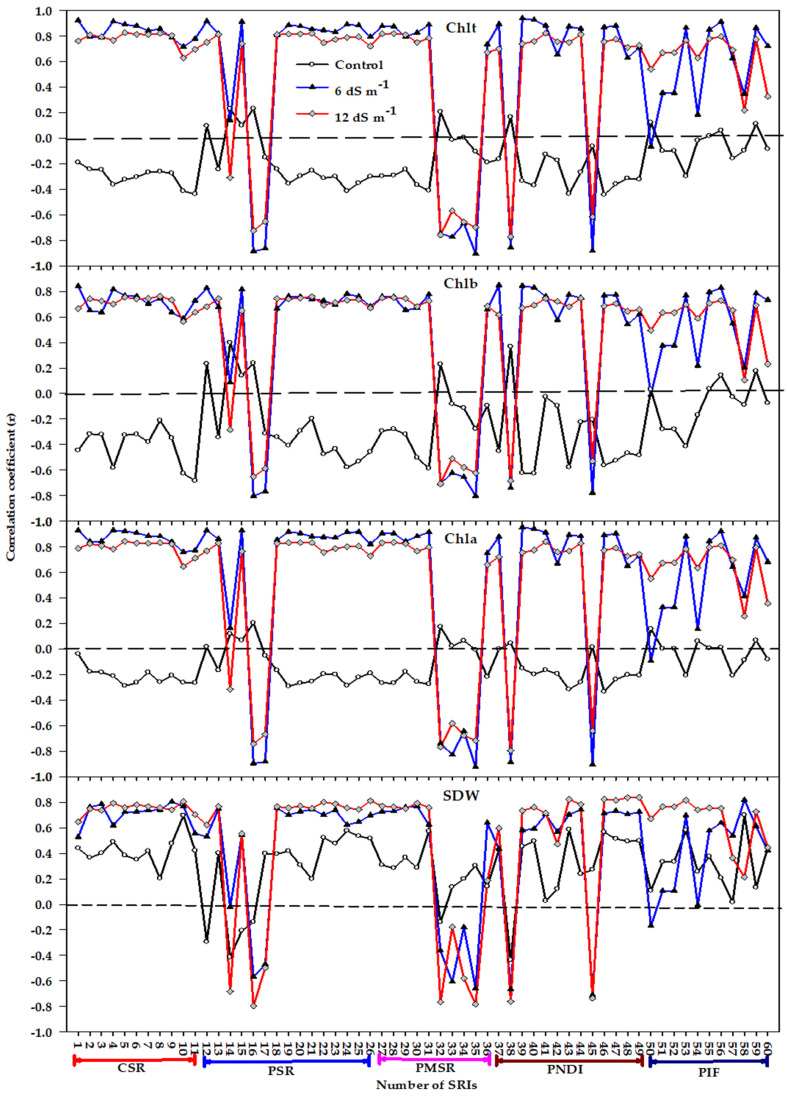
Correlation coefficients (r) for the relationships between different spectral reflectance indices (SRIs) and shoot dry weight (SDW), chlorophyll a (*Chla*), chlorophyll b (*Chlb*), and total chlorophyll (*Chlt*) under each salinity level (*n* = 12). r values ≤ −0.58 and r values ≥ 0.58 are significant at alpha = 0.05. CSR, PSR, PMSR, PND, PIF indicate constructed simple ratio type, published simple ratio type, published modified simple ratio type, published normalized difference type, and published integrated form type, respectively. The name and abbreviation of each number for different types of SRI are listed in Table 1.

**Figure 3 plants-10-00101-f003:**
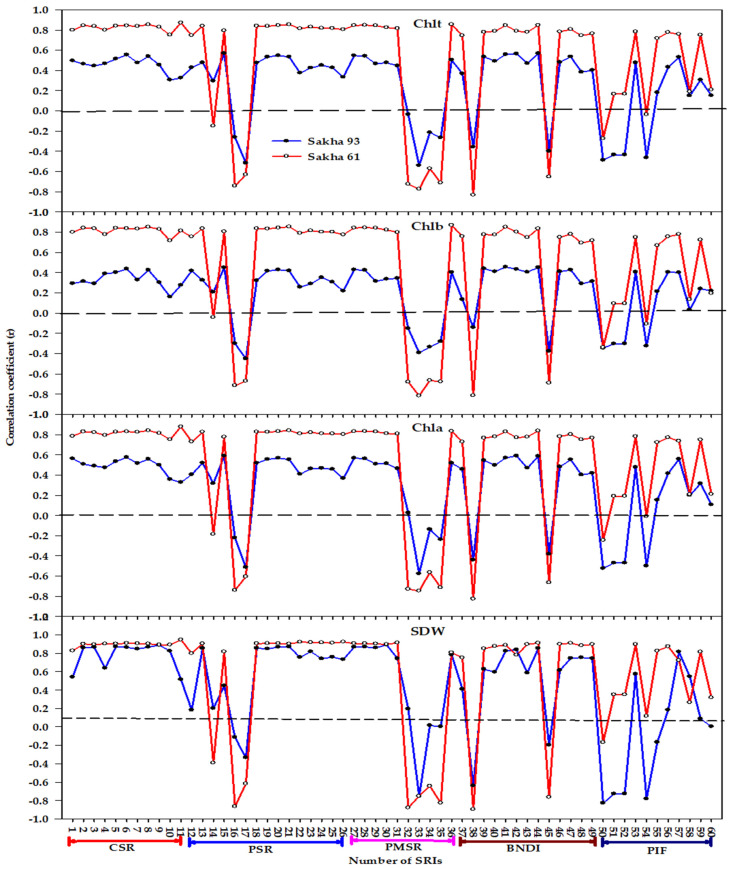
Correlation coefficients (r) for the relationships between different spectral reflectance indices (SRIs) and shoot dry weight (SDW), chlorophyll a (*Chla*), chlorophyll b (*Chlb*), and total chlorophyll (*Chlt*) for each genotype (*n* = 18). r values ≤ −0.47 and r values ≥ 0.47 are significant at alpha = 0.05. CSR, PSR, PMSR, PND, PIF indicate constructed simple ratio type, published simple ratio type, published modified simple ratio type, published normalized difference type, and published integrated form type, respectively. The name and abbreviation of each number for different types of SRI are listed in Table 1.

**Figure 4 plants-10-00101-f004:**
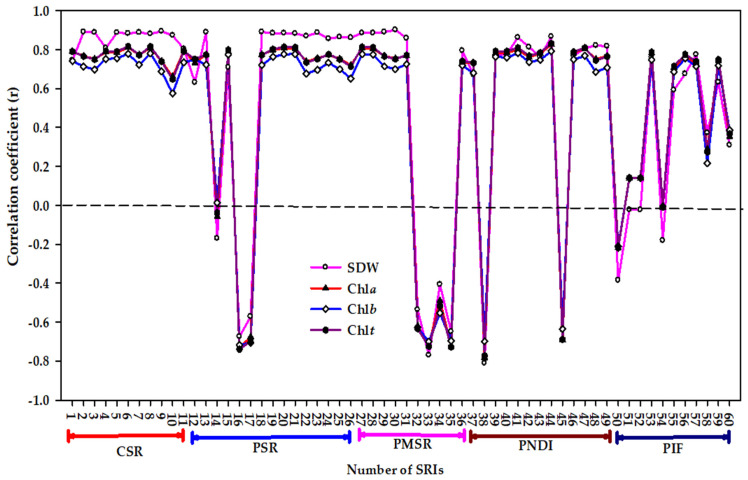
Correlation coefficients (r) for the relationships between different spectral reflectance indices (SRIs) and shoot dry weight (SDW), chlorophyll a (*Chla*), chlorophyll b (*Chlb*), and total chlorophyll (*Chlt*) for pooled data (*n* = 36). r values ≤ −0.35 and r values ≥ 0.35 are significant at alpha = 0.05. CSR, PSR, PMSR, PND, PIF indicate constructed simple ratio type, published simple ratio type, published modified simple ratio type, published normalized difference type, and published integrated form type, respectively. The name and abbreviation of each number for different types of SRI are listed in Table 1.

**Table 1 plants-10-00101-t001:** Full names, abbreviation, and formulas of different types of newly constructed and published vegetation indices used in this study.

NO.	SRIs	Formula
**Constructed simple ratio type (CSR)**		
1	Simple ratio (400 and 478 nm)	R_400_/R_478_
2	Simple ratio (810 and 550 nm)	R_810_/R_550_
3	Simple ratio (970 and 564 nm)	R_970_/R_564_
4	Simple ratio (554 and 572 nm)	R_554_/R_572_
5	Simple ratio (1100 and 700 nm)	R_1100_/R_700_
6	Simple ratio (740 and 710 nm)	R_740_/R_710_
7	Simple ratio (754 and 568 nm)	R_754_/R_568_
8	Simple ratio (762 and 722 nm)	R_762_/R_722_
9	Simple ratio (1250 and 560 nm)	R_1250_/R_560_
10	Simple ratio (1650 and 482 nm)	R_1650_/R_482_
11	Simple ratio (2250 and 2296 nm)	R_2250_/R_2296_
**Published simple ratio type (PSR)**		
12	Simple ratio pigment index-1 (SRPI-1)	R_430_/R_680_
13	Simple ratio pigment index-2 (SRPI-1)	R_750_/R_556_
14	Blue/Green pigment Index-1 (BGI-1)	R_450_/R_550_
15	Blue/Red pigment Index-1 (BRI-1)	R_400_/R_690_
16	Red/green pigment Index-1 (RGI-1)	R_690_/R_550_
17	Red/blue pigment Index (RBI)	R_695_/R_445_
18	Gitelson and merzlyak index 1 (GM-1)	R_750_/R_550_
19	Gitelson and merzlyak index-2 (GM-2)	R_750_/R_700_
20	Ratio analysis of reflectance spectra-a (PARS-a)	R_750_/R_710_
21	Ratio analysis of reflectance spectra-a developed (PARS-a-D)	R_780_/R_720_
22	Ratio analysis of reflectance spectra-c (PARS-c)	R_760_/R_500_
23	Ratio analysis of reflectance spectra-c developed (PARS-c-D)	R_760_/R_515_
24	Pigment Specific Simple ratio-a (PSSR-a)	R_800_/R_675_
25	Pigment-specific simple ratio-b (PSSR-b)	R_800_/R_650_
26	Pigment-specific simple ratio-c (PSSR-c)	R_800_/R_470_
**Published modified simple ratio type (PMSR)**		
27	Red edge chlorophyll index (Cl_Red-edge_)	(R_750_/R_710_) − 1
28	Red edge chlorophyll index –developed (Cl-D_Red-edge_)	(R_760_/R_710_) − 1
29	Green chlorophyll index (Cl_Green_)	(R_800_/R_550_) − 1
30	Ratio analysis of reflectance spectra-a (PARS-b)	R_675_/(R_650_*R_700_)
31	Chlorophyll reflectance index-a (Chl-a)	R_776_ (1/R_673_ − 1)
32	Chlorophyll reflectance index-b (Chl-b)	R_776_ (1/R_625_ − 1/R_673_)
33	Carotenoid reflectance index (CRI)	(1/R_507_ − 1/R_603_ − 0.65*(1/R_530_))*R_776_
34	Anthocyanin (Gitelson) (Ant_Gitelson_)	R_780_ (1/R_550_ − 1/R_700_)
35	Plant senescence reflectance index (PSRI)	(R_680_ − R_500_)/R_750_
36	Red-edge vegetation stress index (RVSI)	0.5 (R_722_ + R_763_) − R_733_
**Published normalized difference type (PND)**		
37	Normalized phaeophytinization index (NPQ)	(R_415_ − R_435_)/(R_415_+R_435_)
38	Normalized phaeophytinization index developed (NPQ-D)	(R_482_ − R_350_)/(R_482_+R_350_)
39	Photochemical reflectance index (PRI)	(R_531_ − R_570_)/(R_531_+R_570_)
40	Photochemical reflectance index developed (PRI-D)	(R_531_ − R_580_)/(R_531_+R_580_)
41	Modified simple ratio of reflectance-1 (MSR-1)	(R_750_ − R_445_)/(R_705_-R_445_)
42	Modified simple ratio of reflectance-2 (MSR-2)	(R_780_ − R_710_)/(R_780_-R_680_)
43	Normalized difference vegetation index (NDVI)	(R_750_ − R_680_)/(R_750_+R_680_)
44	Normalized difference vegetation index developed (NDVI-D)	(R_780_ − R_715_)/(R_780_+R_715_)
45	Structure insensitive pigment index (SIPI)	(R_800_ − R_445_)/(R_800_-R_680_)
46	Pigment specific normalized difference-a (PSND-a)	(R_800_ − R_680_)/(R_800_+R_680_)
47	Pigment specific normalized difference-b (PSND-b)	(R_800_ − R_635_)/(R_800_+R_635_)
48	Pigment specific normalized difference-c (PSND-c)	(R_800_ − R_460_)/(R_800_+R_460_)
49	Pigment specific normalized difference-c developed (PSND-c-D)	(R_800_ − R_482_)/(R_800_+R_482_)
**Published integrated form type (PIF)**		
50	Chlorophyll absorption ratio index (CARI)	[(R_700_ − R_670_) − 0.2 × (R_700_ − R_550_)]
51	Modified chlorophyll absorption ratio index (MCARI)	[(R_700_ − R_670_) − 0.2 × (R_700_ − R_550_)] × (R_700_/R_670_)
52	Transformed chlorophyll absorption reflectance index (TCARI)	3 × ((R_700_ − R_670_) − 0.2 × (R_700_ − R_550_)] × (R_700_/R_670_)
53	Optimized soil-adjusted vegetation index (OSAVI)	1.16(R_800_ − R_670_)/(R_800_ + R_670_ + 0.16)
54	Triangular vegetation index (TVI)	1.2 × (R_700_ − R_550_) − 1.5 × (R_670_ − R_550_) × (R_700_/R_670_)^1/2^]
55	Modified triangular vegetation index (MTVI)	1.2 [1.2(R_800_ − R_550_) − 2.5(R_670_ − R_550_)]
56	Enhanced vegetation index (EVI)	2.5 (R_782_ − R_675_)/(R_782_ + 6 *×* R_675_ − 7.5 *×* R_445_ + 1))
57	Red edge inflection point (REIP)	REIP = 700 + 40 *×* {[(R_670_ + R_780_)/2 − R_700_]/(R_740_ − R_700_)}
58	Salinity and water stress index-1 (SWSI-1)	(R_803_ − R_681_)/root(R_905_ − R_972_)
59	Salinity and water stress index-2 (SWSI-2)	(R_803_ − R_681_)/root(R_1326_ − R_1507_)
60	Salinity and water stress index-3 (SWSI-3)	(R_803_ − R_681_)/root(R_972_ − R_1174_)

**Table 2 plants-10-00101-t002:** Statistical analysis including analysis of variance (degrees of freedom (df), *F*-values, and significance level) of the effect of year, salinity level, genotype, and their interaction on shoot dry weight (SDW), chlorophyll a (*Chla*), chlorophyll b (*Chlb*), and total chlorophyll (*Chlt*) measured at the anthesis growth stage (about 75 days from sowing).

*F*-Values						
Source of Variance		df	SDW (g plant^−1^)	*Chla* (mg g^−1^ Fresh Weight)	*Chlb* (mg g^−1^ Fresh Weight)	*Chlt* (mg g^−1^ Fresh Weight)
Year (Y)		1	119.8 **	32.8 *	905.3 **	88.7 *
Salinity (S)		2	574.9 ***	315.2 ***	73.3 ***	323.1 ***
S*Y		2	0.38 ^ns^	2.65 ^ns^	2.16 ^ns^	0.26 ^ns^
Genotypes (G)		1	119.6 ***	30.3 ***	54.1 ***	37.9 ***
G*Y		1	1.28 ^ns^	0.029 ^ns^	0.91 ^ns^	0.15 ^ns^
G*S		2	19.8 ***	12.3 **	9.59 **	12.1 **
G*S*Y		2	1.04 ^ns^	0.64 ^ns^	0.29 ^ns^	0.57 ^ns^
Mean values of the main factor ± standard deviation						
Salinity	Control	7.70a ± 0.50		2.65a ± 0.22	0.844a ± 0.11	3.49a ± 0.31
	6 dS m^−1^	5.34b ± 0.82		2.34b ± 0.25	0.692b ± 0.14	3.03b ± 0.38
	12 dS m^−1^	3.67c ± 0.91		1.93c ± 0.37	0.580c ± 0.13	2.51c ± 0.50
Genotypes	Sakha 93	6.04a ± 1.55		2.48a ± 0.21	0.784a ± 0.12	3.26a ± 0.34
	Sakha 61	5.10b ± 2.02		2.14b ± 0.47	0.626b ± 0.17	2.76b ± 0.64

*, **, *** Significant at the 0.05, 0.01 and 0.001 probability levels, respectively, and ^ns^: not significant.

**Table 3 plants-10-00101-t003:** F-test for the effect of year (Y), salinity level (S), genotype (G) and their possible interaction on 60 distinct spectral reflectance indices (SRIs) at the anthesis growth stages (about 75 days from sowing). Data are averaged over two seasons.

SRIs	Y		S	S × Y	G	G × Y	G × S	G × S × Y	SRIs	Y		S	S × Y	G	G × Y	G × S	G × S × Y
**Constructed simple ratio type (CSR)**	1	ns	***	ns	***	ns	***	ns	**Published modified simple ratio type (PMSR)**	31	ns	***	ns	***	ns	***	ns
	2	ns	***	ns	***	ns	*	ns		32	ns	*	ns	***	*	**	ns
	3	ns	***	ns	***	ns	*	ns		33	ns	***	ns	**	ns	ns	ns
	4	ns	***	ns	***	ns	***	ns		34	ns	ns	ns	*	ns	ns	ns
	5	ns	***	ns	***	ns	**	ns		35	ns	***	ns	***	*	***	ns
	6	**	***	ns	***	ns	***	ns		36	ns	***	ns	***	ns	ns	ns
	7	ns	***	ns	***	ns	**	ns	**Published normalized difference type (PND)**	37	ns	***	*	***	ns	**	*
	8	ns	***	ns	***	ns	**	ns		38	ns	***	ns	***	ns	***	*
	9	*	***	ns	**	ns	ns	ns		39	ns	***	ns	***	ns	***	ns
	10	ns	***	***	***	ns	ns	ns		40	ns	***	ns	***	ns	***	ns
	11	ns	***	ns	***	ns	**	ns		41	ns	***	ns	***	ns	**	ns
**Published simple ratio type (PSR)**	12	ns	**	ns	***	ns	**	ns		42	ns	***	ns	**	ns	ns	ns
	13	ns	***	ns	***	ns	**	ns		43	ns	***	ns	***	*	***	ns
	14	ns	ns	ns	ns	ns	ns	ns		44	*	***	ns	***	ns	***	ns
	15	ns	***	ns	***	ns	**	ns		45	ns	**	ns	***	ns	**	ns
	16	ns	***	ns	***	*	***	ns		46	ns	***	ns	***	*	***	ns
	17	ns	***	ns	***	ns	*	ns		47	ns	***	ns	***	*	***	ns
	18	ns	***	ns	***	ns	*	ns		48	ns	***	ns	***	ns	**	ns
	19	*	***	ns	***	ns	***	ns		49	ns	***	ns	***	ns	**	ns
	20	**	***	ns	***	ns	***	ns	**Published integrated form type (PIF)**	50	ns	*	ns	ns	ns	ns	ns
	21	ns	***	ns	***	ns	**	ns		51	ns	ns	ns	**	ns	*	ns
	22	ns	***	*	***	ns	**	ns		52	ns	ns	ns	**	ns	*	ns
	23	ns	***	ns	***	ns	**	ns		53	ns	***	ns	***	*	***	ns
	24	ns	***	ns	***	ns	**	ns		54	ns	ns	ns	*	ns	*	ns
	25	ns	***	*	***	ns	***	ns		55	ns	**	ns	***	*	***	ns
	26	ns	***	**	***	ns	*	ns		56	ns	***	ns	***	*	***	*
	27	**	***	ns	***	ns	***	ns		57	ns	***	ns	**	ns	ns	ns
	28	*	***	ns	***	ns	***	ns		58	ns	ns	ns	ns	ns	ns	ns
	29	ns	***	ns	***	ns	*	ns		59	ns	***	ns	***	ns	**	ns
	30	*	***	ns	***	ns	***	ns		60	ns	ns	ns	**	ns	ns	ns

*, **, *** Significant at the 0.05, 0.01 and 0.001 probability levels, respectively, and ns: not significant. The name and abbreviation of each number for different types of SRI are listed in Table 1.

**Table 4 plants-10-00101-t004:** Comparison of the mean values of 60 spectral reflectance indices (SRIs) among the two genotypes and three salinity levels (data are averaged over two seasons).

SRIs		Genotypes			Salinity Levels			SRIs		Genotypes			Salinity Levels		
		S93	S61	C		6 dS m^−1^	12 dS m^−1^			S93	S61	C		6 dS m^−1^	12 dS m^−1^
**Constructed simple ratio type (CSR)**	1	0.79a	0.64b	0.84a		0.67b	0.62b	**Published modified simple ratio type (PMSR)**	31	10.56a	6.18b	14.32a		5.95b	4.84b
	2	5.62a	4.25b	7.01a		4.25b	3.55c		32	−1.82b	−0.77a	−1.90b		−0.95a	−1.03a
	3	5.81a	4.33b	7.39a		4.25b	3.57c		33	−2.04b	−1.24a	−2.90b		−1.17a	−0.86a
	4	1.19a	1.08b	1.25a		1.09b	1.07c		34	0.40b	0.70a	0.31a		0.69a	0.65a
	5	5.11a	3.49b	6.49a		3.51b	2.90c		35	0.009b	0.11a	0.004b		0.084a	0.091a
	6	2.50a	1.94b	2.92a		1.99b	1.75c		36	0.032a	0.021b	0.039a		0.024b	0.016c
	7	5.95a	4.28b	7.64a		4.23b	3.48c	**Published normalized difference type (PND)**	37	−0.057a	−0.088b	−0.046a		−0.081b	−0.090b
	8	1.92a	1.60b	2.14a		1.66b	1.49c		38	−0.014b	0.14a	−0.091b		0.11a	0.17a
	9	4.87a	3.76b	6.09a		3.68b	3.17c		39	0.023a	−0.028b	0.045a		−0.024b	−0.029b
	10	6.23a	4.98b	7.84a		4.69b	4.28c		40	0.083a	−0.001b	0.119a		0.007b	−0.002b
	11	1.17a	1.13b	1.18a		1.14b	1.12b		41	4.87a	3.28b	6.08a		3.40b	2.75c
**Published simple ratio type (PSR)**	12	0.83a	0.56b	0.89a		0.61b	0.60b		42	0.74a	0.68b	0.79a		0.70b	0.65c
	13	5.13a	3.85b	6.38a		3.84b	3.24c		43	0.80a	0.57b	0.85a		0.62b	0.59b
	14	0.48a	0.47a	0.47a		0.47a	0.47a		44	0.43a	0.30b	0.49a		0.33b	0.27c
	15	0.57a	0.38b	0.64a		0.41b	0.38b		45	1.02b	1.24a	1.01b		1.17a	1.22a
	16	1.00a	0.65b	0.94a		0.94a	0.60b		46	0.81a	0.60b	0.86a		0.65b	0.61b
	17	2.47a	1.79b	2.40a		2.36a	1.64b		47	0.79a	0.59b	0.86a		0.63b	0.59b
	18	5.10a	3.87b	6.32a		3.87b	3.27c		48	0.84a	0.76b	0.87a		0.78b	0.74c
	19	4.73a	3.22b	6.03a		3.22b	2.66c		49	0.84a	0.74b	0.87a		0.76b	0.73c
	20	2.85a	2.14b	3.38a		2.21b	1.90c	**Published integrated form type (PIF)**	50	0.083a	0.074a	0.094a		0.080ab	0.062b
	21	2.11a	1.73b	2.37a		1.79b	1.59c		51	0.186a	0.132b	0.145a		0.149a	0.182a
	22	10.91a	7.67b	14.14a		7.48b	6.25b		52	0.558a	0.395b	0.436a		0.448a	0.547a
	23	8.56a	6.16b	11.1a		6.02b	4.96c		53	0.803a	0.599b	0.845a		0.646b	0.613b
	24	11.23a	6.82b	14.81a		6.67b	5.60b		54	0.186a	0.147b	0.142b		0.163ab	0.195a
	25	11.20a	7.02b	15.55a		6.49b	5.28b		55	1.103a	0.778b	1.079a		0.880b	0.862b
	26	12.08a	8.80b	15.38a		8.66b	7.29c		56	0.999a	0.654b	1.023a		0.736b	0.720b
	27	1.85a	1.14b	2.38a		1.21b	0.99c		57	722.31a	720.01b	723.98a		720.61b	718.88c
	28	2.01a	1.25b	2.60a		1.32b	0.97c		58	3.06a	2.84a	3.13a		2.92a	2.80a
	29	4.60a	3.23b	5.99a		3.23b	2.53c		59	1.17a	0.99b	1.17a		1.05b	1.03b
	30	6.81a	5.09b	9.53a		4.66b	3.65c		60	2.11a	1.86b	2.06a		1.98b	1.91b

Means followed by the same letter are not significantly different from one another based on Fisher’s least significant difference (LSD) test at *p* ≤ 0.05. S93 and S61 indicate genotypes Sakha 93 and Sakha 61, respectively. C indicate control. The name and abbreviation of each number for different type of SRIs are listed in Table 1.

**Table 5 plants-10-00101-t005:** Extraction of the most influential spectral reflectance indices (SRIs) from each type of SRIs accounting for the major variation for shoot dry weight (SDW) and content of chlorophyll a (*Chla*), chlorophyll b (*Chlb*), and total chlorophyll (*Chlt*) using stepwise multiple linear regression (SMLR) analysis and based on pooled data (*n* = 36).

SRIs Groups	Measured Variables	Influential SRI	Best Fitted Equation	Model R^2^	Model RMSE
Constructed simple ratio type (CSR)	SDW	SRI_(1250,560)_	SDW = 0.909 +1.08 (SRI_(1250,560)_)	0.80	0.836
	*Chla*	SRI_(740,710)_	*Chla* = 1.139 + 0.526 (SRI_(740,710)_)	0.66	0.242
	*Chlb*	SRI_(700,1100)_	*Chlb* = 0.974 − 0.928 (SRI_(700,1100)_)	0.62	0.105
	*Chlt*	SRI_(740,710)_	*Chlt* = 1.358 + 0.732 SRI_(740,710)_	0.67	0.331
Published simple ration type (PSR)	SDW	GMI	SDW = 1.006 + 1.017 (GMI)	0.79	0.846
	*Chla*	PARS-a	*Chla* = 1.268 + 0.416 (PARS-a)	0.65	0.244
	*Chlb*	PARS-D-a	*Chlb* = 0.114 + 0.308 (PARSD-a)	0.60	0.107
	*Chlt*	PARS-a	*Chlt* = 1.566 + 0.579 (PARS-a)	0.66	0.334
Published modified simple ratio type (PMSR)	SDW	PARS-b	SDW = 2.258 + 0.557 (PARS-b)	0.81	0.806
	*Chla*	CI_Red-edge_	*Chla* = 1.684 + 0.416 (CI_Red-edge_)	0.65	0.244
	*Chlb*	CI_Red-edge_, PARS-b	*Chlb* = 0.50 + 0.43 (CI_Red-edge_) − 0.007 (PARS-b)	0.68	0.098
	*Chlt*	CI_Red-edge_, PARS-b	*Chlt* = 2.23 + 1.22 (CI_Red-edge_) − 0.18 (PARS-b)	0.70	0.318
Published normalized difference type (PND)	SDW	NDVI-D	SDW = 0.950+12.718 (NDV3-D)	0.75	0.928
	*Chla*	NDVI-D	*Chla* = 1.326 + 2.701 (NDVI-D)	0.68	0.234
	*Chlb*	NDVI-D	*Chlb* = 0.320 + 1.06 (NDVI-D)	0.63	0.103
	*Chlt*	NDVI-D	*Chlt* = 1.646 + 3.76 (NDVI-D)	0.69	0.318
Published integrated form type (PIF)	SDW	OSAVI, REIP	SDW = −211.45 + 4.48 (OSAVI) + 0.30 (REIP)	0.69	1.051
	*Chla*	TCARI, OSAVI	*Chla* = 1.076 − 0.602 (TCARI) + 2.165 (OSAVI)	0.68	0.238
	*Chlb*	EVI-1, REIP	*Chlb* = −15.074 + 0.316 (EVI-1) + 0.002 (REIP)	0.65	0.102
	*Chlt*	OSAVI, REIP	*Chlt* = −49.277 + 1.72 (OSAVI) +0.007 (REIP)	0.69	0.325

The full name of the abbreviation of each spectral reflectance indices are listed in Table 1.

**Table 6 plants-10-00101-t006:** Function of linear validations between the observed and predicted values, coefficient of determination (R^2^), and root mean square error (RMSE) of linear regression models based on an individual selected spectral index (Table 5). These models were calibrated using a dataset of 2 seasons. Subsequently, the equations of calibration of distinct models (Table 5) were used to predict the shoot dry weight (SDW) and content of chlorophyll a (*Chla*), chlorophyll b (*Chlb*), and total chlorophyll (*Chlt*) for each salinity level.

SRI Type	Measured Variables	Control			Moderate Salinity Level (6 dS m^−1^)			High Salinity Level (12 dS m^−1^)		
		Equation	R^2^	RMSE	Equation	R^2^	RMSE	Equation	R^2^	RMSE
Constructed simple ratio type (CSR)	SDW	y = 0.7885x + 1.411	0.23	0.732	y = 1.1424x − 1.208	0.65	0.809	y = 0.476x + 2.587	0.55	0.889
	*Chla*	y = −0.0549x + 2.789	0.04	0.230	y = 1.0082x − 0.151	0.67	0.212	y = 0.5552x + 0.995	0.68	0.250
	*Chlb*	y = −0.0432x + 0.866	0.11	0.113	y = 0.6707x + 0.199	0.58	0.095	y = 0.6103x + 0.269	0.53	0.098
	*Chlt*	y = −0.0564x + 3.649	0.11	0.317	y = 1.0317x − 0.267	0.66	0.272	y = 0.628x + 1.144	0.64	0.357
Published simple ration type (PSR)	SDW	y = 0.5635x + 3.0961	0.16	0.708	y = 1.1356x − 1.121	0.57	0.880	y = 0.6262x + 2.034	0.59	0.868
	*Chla*	y = −0.1091x + 2.963	0.07	0.248	y = 1.0194x − 0.199	0.82	0.190	y = 0.398x + 1.290	0.69	0.268
	*Chlb*	y = −0.0754x + 0.911	0.04	0.122	y = 0.5676x + 0.273	0.55	0.097	y = 0.3971x + 0.374	0.58	0.091
	*Chlt*	y = −0.1208x + 3.946	0.09	0.350	y = 0.8905x + 0.145	0.77	0.261	y = 0.4049x + 1.649	0.67	0.353
Published modified simple ratio type (PMSR)	SDW	y = 0.3545x + 4.833	0.08	0.655	y = 1.0647x − 0.831	0.59	0.852	y = 0.5072x + 2.430	0.63	0.829
	*Chla*	y = −0.1091x + 2.963	0.07	0.248	y = 1.0194x − 0.199	0.82	0.190	y = 0.398x + 1.290	0.69	0.268
	*Chlb*	y = 0.1155x + 0.727	0.06	0.109	y = 0.8061x + 0.118	0.68	0.083	y = 0.6485x + 0.239	0.61	0.087
	*Chlt*	y = −0.0231x + 3.543	0.00	0.338	y = 1.0921x − 0.424	0.80	0.248	y = 0.5797x + 1.230	0.71	0.321
Published normalized difference type (PND)	SDW	y = 0.1511x + 6.037	0.06	0.705	y = 1.3114x − 1.892	0.55	0.992	y = 0.9458x + 0.929	0.62	0.979
	*Chla*	y = −0.0796x + 2.864	0.07	0.235	y = 1.109x − 0.388	0.88	0.194	y = 0.5149x + 1.064	0.69	0.249
	*Chlb*	y = −0.0529x + 0.886	0.05	0.115	y = 0.6297x + 0.231	0.66	0.096	y = 0.5149x + 0.309	0.56	0.089
	*Chlt*	y = −0.0811x + 3.778	0.07	0.329	y = 0.9726x − 0.075	0.73	0.264	y = 0.524x + 1.350	0.66	0.331
Published integrated form type (PIF)	SDW	y = 0.1053x + 6.239	0.03	0.830	y = 1.2462x − 1.491	0.47	1.076	y = 0.9116x + 1.153	0.56	1.091
	*Chla*	y = −0.038x + 2.743	0.03	0.222	y = 1.1193x − 0.415	0.78	0.198	y = 0.5564x + 0.999	0.64	1.504
	*Chlb*	y = 0.0103x + 0.819	0.00	0.110	y = 0.7442x + 0.150	0.62	0.091	y = 0.6731x + 0.233	0.61	0.091
	*Chlt*	y = −0.0648x + 3.685	0.06	0.324	y = 1.0211x − 0.218	0.73	0.274	y = 0.6243x + 1.131	0.68	0.331

**Table 7 plants-10-00101-t007:** Function of linear validations between the observed and predicted values, coefficient of determination (R^2^), and root mean square error (RMSE) of linear regression models based on an individual selected spectral index (Table 5). These models were calibrated using a dataset of 2 seasons. Subsequently, the equations of calibration of distinct models (Table 5) were used to predict the shoot dry weight (SDW) and content of chlorophyll a (*Chla*), chlorophyll b (*Chlb*), and total chlorophyll (*Chlt*) for each genotype.

SRI Type	Measured Variables	Salt-Tolerant Genotype Sakha 93			Salt-Sensitive Genotype Sakha 61		
		Equation	R^2^	RMSE	Equation	R^2^	RMSE
Constructed simple ratio type (CSR)	SDW	y = 0.6537x + 2.836	0.84	1.388	y = 0.725x + 1.271	0.80	0.907
	*Chla*	y = 0.4303x + 1.403	0.30	0.203	y = 0.6552x + 0.745	0.69	0.257
	*Chlb*	y = 0.2178x + 0.607	0.22	0.103	y = 0.6688x + 0.214	0.64	0.102
	*Chlt*	y = 0.3355x + 2.175	0.34	0.271	y = 0.66x + 0.931	0.67	0.357
Published simple ration type (PSR)	SDW	y = 0.7019x + 0.858	0.73	1.223	y = 0.7786x + 0.976	0.82	0.850
	*Chla*	y = 0.537x + 1.122	0.32	0.214	y = 0.6259x + 0.824	0.69	0.259
	*Chlb*	y = 0.328x + 0.507	0.18	0.115	y = 0.6721x + 0.226	0.73	0.091
	*Chlt*	y = 0.5062x + 1.564	0.30	0.310	y = 0.6569x + 0.994	0.71	0.338
Published modified simple ratio type (PMSR)	SDW	y = 0.7391x + 1.585	0.79	0.688	y = 0.8311x + 0.851	0.81	0.868
	*Chla*	y = 0.537x + 1.122	0.32	0.214	y = 0.6259x + 0.824	0.69	0.259
	*Chlb*	y = 0.374x + 0.500	0.32	0.096	y = 0.6289x + 0.224	0.72	0.091
	*Chlt*	y = 0.5043x + 1.645	0.34	0.276	y = 0.6235x + 1.012	0.73	0.331
Published normalized difference type (PND)	SDW	y = 0.5115x + 3.273	0.73	0.930	y = 0.7846x + 0.775	0.83	0.873
	*Chla*	y = 0.479x + 1.288	0.34	0.197	y = 0.6514x + 0.747	0.70	0.259
	*Chlb*	y = 0.2923x + 0.542	0.20	0.107	y = 0.6961x + 0.203	0.70	0.094
	*Chlt*	y = 0.4536x + 1.767	0.32	0.286	y = 0.6805x + 0.898	0.72	0.332
Published integrated form type (PIF)	SDW	y = 0.4501x + 3.650	0.68	1.007	y = 0.7153x + 1.121	0.77	1.006
	*Chla*	y = 0.453x + 1.356	0.37	0.189	y = 0.6467x + 0.753	0.68	0.262
	*Chlb*	y = 0.2622x + 0.579	0.25	0.101	y = 0.6766x + 0.202	0.69	0.094
	*Chlt*	y = 0.3802x + 2.029	0.32	0.276	y = 0.6676x + 0.911	0.70	0.342

**Table 8 plants-10-00101-t008:** Function of linear validations between the observed and predicted values, coefficient of determination (R^2^), and root mean square error (RMSE) of linear regression models based on an individual selected spectral index (Table 5). These models were calibrated using a dataset of 2 seasons. Subsequently, the equations of calibration of distinct models (Table 5) were used to predict the shoot dry weight (SDW) and content of chlorophyll a (*Chla*), chlorophyll b (*Chlb*), and total chlorophyll (*Chlt*) for each season.

SRI Type	Measured Variables	First Season			Second Season		
		Equation	R^2^	RMSE	Equation	R^2^	RMSE
Constructed simple ratio type (CSR)	SDW	y = 0.9072x + 0.326	0.82	0.777	y = 0.7325x + 1.609	0.80	0.847
	*Chla*	y = 0.6009x + 0.996	0.69	0.256	y = 0.8293x + 0.302	0.79	0.203
	*Chlb*	y = 0.7281x + 0.250	0.74	0.102	y = 0.7802x + 0.093	0.84	0.102
	*Chlt*	y = 0.6186x + 1.270	0.68	0.351	y = 0.8533x + 0.288	0.84	0.279
Published simple ration type (PSR)	SDW	y = 0.8832x + 0.437	0.84	0.754	y = 0.7534x + 1.533	0.78	0.886
	*Chla*	y = 0.6209x + 0.932	0.64	0.262	y = 0.7465x + 0.518	0.74	0.211
	*Chlb*	y = 0.7786x + 0.212	0.70	0.103	y = 0.6996x + 0.161	0.80	0.105
	*Chlt*	y = 0.6751x + 1.090	0.67	0.348	y = 0.3955x + 1.421	0.76	0.595
Published modified simple ratio type (PMSR)	SDW	y = 0.8981x + 0.372	0.85	0.712	y = 0.7721x + 1.412	0.79	0.849
	*Chla*	y = 0.6209x + 0.932	0.64	0.262	y = 0.7465x + 0.518	0.74	0.211
	*Chlb*	y = 0.73x + 0.230	0.68	0.096	y = 0.8121x + 0.088	0.82	0.092
	*Chlt*	y = 0.6396x + 1.158	0.66	0.377	y = 0.8358x + 0.395	0.81	0.269
Published normalized difference type (PND)	SDW	y = 0.7573x + 1.126	0.81	0.836	y = 0.8031x + 1.341	0.75	0.963
	*Chla*	y = 0.6154x + 0.949	0.66	0.256	y = 0.8245x + 0.329	0.80	0.194
	*Chlb*	y = 0.7773x + 0.215	0.73	0.101	y = 0.7583x + 0.114	0.84	0.100
	*Chlt*	y = 0.671x + 1.107	0.70	0.378	y = 0.8217x + 0.400	0.83	0.278
Published integrated form type (PIF)	SDW	y = 0.7105x + 1.3684	0.72	0.985	y = 0.7319x + 1.745	0.71	1.027
	*Chla*	y = 0.5726x + 1.047	0.65	0.261	y = 0.8698x + 0.218	0.81	0.190
	*Chlb*	y = 0.7721x + 0.209	0.69	0.099	y = 0.7603x + 0.121	0.82	0.096
	*Chlt*	y = 0.6558x + 1.147	0.67	0.346	y = 0.8282x + 0.383	0.84	0.271

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
