# Peer review of "Use of Hyperspectral Reflectance Sensing for Assessing Growth and Chlorophyll Content of Spring Wheat Grown under Simulated Saline Field Conditions"

_plants, 2021, doi:10.3390/plants10010101_

Round 1

Reviewer 1 Report

I have enjoyed reading it.

Author Response

Reviewer #1

Manuscript ID: plants-1057945

Title: Use of Hyperspectral Reflectance Sensing for Assessing Growth and Chlorophyll Content of Spring Wheat Grown Under Simulated Saline Field Conditions

Recommendation:

Accepted with revisions

Response: We greatly appreciate your critical observations as well as your constructive and helpful comments. We hope that we could address your questions/comments by the explanations and revisions made in the manuscript. We believe that the manuscript is substantially improved after making the suggested revisions.

General comments:

This manuscript describes and interprets the results from two experiments conducted in 2017/18 and 2018/19 using two contrasting saline tolerant spring wheat genotypes grown under three saline levels in the fields. 60 different spectral reflectance indicators (SRIs) were derived and used to relate to shoot dry matter per plant and concentrations of chlorophylls a, b and the combination of chlorophyll a and b. Out of 60 SRIs, 11 of the SRIs were original in this study and 49 other SRIs were derivatives from published literature. It was thorough and well written even though it was a bit long, especially the introduction. The experimental designs and analysis methods were sound, but the details were inadequate of how the spectral reflectance measurements were done in the fields. Interpretations of the data were credible, and the findings have explorable implications, particularly in these dry areas where irrigation is necessary for growing crops.

As described in the results that shoot dry matter and chlorophyll contents differed significantly, but the measured SRIs were not significantly different between the two years, this suggested that these SRIs had limited values because their discriminating ability was weak.  This may have pointed out that the starting time of measurement at 75 days from sowing was too late or more measurements were needed in the late stages of the crop growth and development.

Response: Many thanks for your valuable comments and suggestion about our manuscript. The spectral reflectance measurements have been explained in details under sub-title Hyperspectral Reflectance Measurements.  

Minor comments:

  1. Introduction

PG3 L104 “VIS” has not been given the full meaning before it was mentioned. If it is meant to be “visible”, then give the full word first.

Response: The full names of the abbreviation of “VIS, NIR, and SWIR” are already mentioned in the Lines 96, 99, and 101, respectively.   

PG4 L163 Give the full spelling of the words used for the acronym “SMLR”. Is it “stepwise multiple linear regression”?

Response: Many thanks for this remind. The full name of the acronym “SMLR” has been written (Line 164). 

  1. Materials and Methods

PG4 L201 Give the number of times the irrigation was applied in each of the growing seasons and the amount of water that was given at each application, so readers can work out the total amount of irrigation applied. When the irrigation was needed and applied, what was the soil water content relative to the field capacity?

Response: Many thanks for this remind. All information about the time of irrigation as well as the amount of irrigation water has been mentioned in detail “Lines 201-203).  

PG5 L215 Spectral reflectance measurement started 75 days from sowing. It is also useful for readers to know the physiological stage of the wheat crop at this point in each year.

Response: Many thanks for this remind. The physiological stage of wheat for measurements of spectral reflectance has been provided “Line 215-216).  

PG5 L216 “…at distinct positions…”. Tell readers what these distinct positions are and why they are distinct.

Response: This means that the spectral reflectance measurements were taken at different places on the internal rows in each subplot (five random places). This information has been provided in the lines 216 and 217. 

PG5 L230. 20 plants were randomly collected from each subplot within the spectral collection area. Can these plants be used to scale up to work out the total crop dry matter production per hectare? If so, give the plant population in the subplot (i.e., what the numbers plants/shoots per square metre?).  Readers can then work out the total dry matter productivity per hectare using the shoot dry matter per plant in part two of Table 2 at PG9 Ls315 - 319.

Response: In this study, the shoot dry weight was calculated based on individual plant not based on area. If we need to calculate the total dry weight based on area, we should sampled plants from specific area (approximately 0.15 m2).      

PG5 L231   What is the size of “the spectral collection area” in metre squares?

Response: As we mentioned in the lines “226–228”. Because the optical fiber probe of the device had a 25° field of view, the probe could detect the spectral reflectance from a circular area of canopy with a 23.0 cm diameter when it was held vertically at approximately 0.8 m above the canopy in the nadir orientation.      

PG5 L232 “SDW” Give the full spelling for the initial of the acronym first. Was it Shoot Dry Weight (SDW)? Also, need to explain to readers what was included in the shoot.

Response: Many thanks for this remind. The full name of SDW has been written (Line 235). Shoot of plants means the different parts of above-ground plant (stem, leaves, and spike). This information has been provided in line “223”.

  1. Results

PG8 L304 In Figure 1, the shoot dry matter (SDW) was not correctly labelled since the “TDM” was used in the Figure 1.  It looks like that SDM represented total dry matter (TDM). This was why it was necessary to define what parts were made of shoots (see above comment on SDM).

Response: Many thanks for this remind. The TDM has been changed to SDW in the Figure 1.

PG10 L346 In Table 3, “…at anthesis growth stage…”, it is also good to give the number days from sowing at this stage in each year.

Response: The number of days from sowing for this growth stage has been provided in Tables 2 and 3.

PG13 L371 In Figure 3, the two genotypes may be further distinguished by two different symbols (for example, open squares and open circles, respectively).

Response: The two genotypes have been distinguished by two different symbols in Figure 3.

PG13 L385 In Figure 4, the four different variables should be better distinguished by four different symbols (for example, open and solid squares, open and solid circles, respectively).

Response: The four different variables have been distinguished by four different symbols in Figure 4.

PG15 L419 In footnote of Table 5, “indicx” should be index or indices.

Response: Indich has been changed into indices

PG15 Ls-423-424 “… the distinct predictive models for each SRIs type failed to predict any variable under the control treatment”. The accuracy of the predictions cannot be only measured by the variance accounted for or coefficient of determination (i.e. the magnitude of R2). It should also take the values of RMSE into account.

Response: The differences value of RMSE between different predictive models of SRIs type for each variable not also big different as well as they were associated with the value of R2. For example, the values of RMSE for SDW, Chla, Chlb, and Chlt were ranged from 0.655 to 0.830, from 0.222 to 0.248, from 0.109 to 0.122, and from 0.317 to 0.350, respectively.   

Also consider Tables 6-8 to be made supplementary materials because all this information has already been contained in the predictive models which were derived from all the 36 data points from all treatments in two years as shown in Table 5.

Response: The Tables 6-8 is very important to include in the manuscript because they related to predict the measured variables under different conditions (salinity levels, genotypes, and seasons)

  1. Discussion

PG21 L581 Refence styles of “Sims and Gamon [77], Gitelson et al. [39], and Lu et al. [23] were changed.

Response: This is because these references are mentioned at the beginning of the sentence and not at the end

  1. References

PG22 L654 “2017a” in reference 1, the “a” should be deleted.

Response: There are two references for El-Hendawy et al. in 2017 (Ref No. 1 and No. 7). Therefore the both references were distinguished by a and b.

PG24 L743 “2019a” in reference 36, the “a” should be deleted.

Response: There are also four references for El-Hendawy et al. in 2019 (Ref No. 24, 36, 41, and 54). Therefore these references were distinguished by a, b, c, and d.

PG25 L814 “2017a” in reference 65, the “a” should be deleted.

Response: This reference is repeated and has been deleted.

Reviewer 2 Report

The present study is interesting and important from theoretical point of view. Since the variety of vegetation indices is large, the attempt to treat statistically the relationships between different growing conditions and plat properties presented through some vegetation index is deserving interest.

I have some questions and comments to the authors:

  1. Which is the software package used?
  2. 2. The authors mention "experimental design" as an important part of the performance of the study. Please, specify the design and describe it in more details in order to better understand its role in the work. Does the design contribute in carrying out the regression analysis? Or in determination of the mixed interactions (so called 2FI models?
  3. Would it be possible to present not only ANOVA tables but some plots for fitting experimental vs. calculated values?

Author Response

Reviewer #2

The present study is interesting and important from theoretical point of view. Since the variety of vegetation indices is large, the attempt to treat statistically the relationships between different growing conditions and plat properties presented through some vegetation index is deserving interest.

Response: We greatly appreciate your critical observations as well as your constructive and helpful comments. We hope that we could address your questions/comments by the explanations and revisions made in the manuscript. We believe that the manuscript is substantially improved after making the suggested revisions.

I have some questions and comments to the authors:

  1. Which is the software package used?

Response: Many thanks for this remind. The different statistical analysis and plotting were performed using R software v. 3.6.1 (R Core Team 2017) and Sigma Plot for Windows (Version 12.0, SPSS, Chicago, IL, USA). This information has been provided in the text (Lines 285-287). 

  1. The authors mention "experimental design" as an important part of the performance of the study. Please, specify the design and describe it in more details in order to better understand its role in the work. Does the design contribute in carrying out the regression analysis? Or in determination of the mixed interactions (so called 2FI models?

Response: Many thanks for this comment. The experimental design has been provided in Lines 204-205 under sub-title “Salinity Treatments, Experimental Design, and Agronomic Practices” as well as in Lines 271-273 under sub-title “Data Analysis”

  1. Would it be possible to present not only ANOVA tables but some plots for fitting experimental vs. calculated values?

Response: Comparison of the mean values of measured variables among the two genotypes and three salinity levels has been provided in Table 2. Comparison of the mean values of sixty spectral reflectance indices (SRIs) among the two genotypes and three salinity levels has been provided in Table 4.
